# Impact of Prior Pulsed Electric Field and Chitooligosaccharide Treatment on Trypsin Activity and Quality Changes in Whole and Beheaded Harpiosquillid Mantis Shrimp during Storage in Iced Water

**DOI:** 10.3390/foods13010028

**Published:** 2023-12-20

**Authors:** Mallikarjun Chanchi Prashanthkumar, Wattana Temdee, Ajay Mittal, Watcharapol Suyapoh, Peerapon Sornying, Suriya Palamae, Jirayu Bautong, Bin Zhang, Hui Hong, Soottawat Benjakul

**Affiliations:** 1International Center of Excellence in Seafood Science and Innovation, Faculty of Agro-Industry, Prince of Songkla University, Hat Yai 90110, Thailand; mallikarjun.chanchi@gmail.com (M.C.P.); wattanatemdee@gmail.com (W.T.); 6411030001@email.psu.ac.th (A.M.); suriya.pal@psu.ac.th (S.P.); jirayu.b@psu.ac.th (J.B.); 2Veterinary Pathology Unit, Department of Veterinary Science, Faculty of Veterinary Science, Prince of Songkla University, Hat Yai 90110, Thailand; watcharapol.s@psu.ac.th (W.S.); peerapon.s@psu.ac.th (P.S.); 3Key Laboratory of Health Risk Factors for Seafood of Zhejiang Province, College of Food Science and Pharmacy, Zhejiang Ocean University, Zhoushan 316022, China; zhangbin@zjou.edu.cn; 4Beijing Laboratory for Food Quality and Safety, College of Food Science and Nutritional Engineering, China Agricultural University, Beijing 100083, China; hhong@cau.edu.cn

**Keywords:** chitooligosaccharide, HMS, PEF, trypsin activity

## Abstract

Harpiosquillid mantis shrimp (*Harpiosquilla raphidea*) (HMS) without and with beheading pretreated with pulsed electric field (PEF) (15 kV/cm, 800 pulses, 5 min) were soaked in chitooligosaccharide (COS) solution at varying concentrations (0, 1 and 2%, *w*/*v*) for 20 min and stored for 3 days in iced water. Changes in the trypsin activity, color, texture, protein pattern, TCA soluble peptide content, histological images, protein secondary structure and microbial load were monitored during the storage. The beheaded HMS pretreated with PEF and soaked in 2% COS solution showed the maximum efficacy in inhibiting trypsin activity and proteolysis, thus retaining muscle proteins, especially myosin heavy chain, actin and troponin T as well as shear force up to day 3. Pronounced muscle destruction in the whole HMS was displayed by a decreased mean grey index and fiber gapping. Such changes were lowered by the beheading and PEF/2% COS treatment (2% COS-BH). Nevertheless, no marked change in the secondary structure including α-helix, β-sheets, β-turns and random coil were observed among any of the samples. The microbiological analysis revealed that the total viable count (TVC) was below 6 log CFU/g till day 2 in all samples. Nonetheless, the 2% COS-BH sample had the lowest psychrophilic bacterial count and *Enterobacteriaceae* count at day 3, compared to the others. Thus, the combination of the prior PEF and 2% COS treatment of beheaded HMS could effectively inhibit proteases, retard the microbial growth and maintain the quality of HMS stored in iced water.

## 1. Introduction

Harpiosquillid mantis shrimp (HMS) (*Harpiosquilla raphidea*) is a high-valued crustacean commonly consumed in Thailand, Myanmar, Indonesia, Singapore, Vietnam and Malaysia [1,2]. After death, HMS is highly sensitive to rapid autolysis, associated with meat softening during storage [2]. Generally, autolysis is mediated by alkaline, neutral or acidic proteases, which are responsible for textural changes in shrimp muscle. Endogenous enzymes in the hepatopancreas majorly cause softening and textural degradation of shrimp muscles. Chanchi Prashanthkumar et al. [3] reported that serine proteases, mainly trypsin, are mainly involved in the muscle deterioration and quality loss of HMS, respectively. Trypsin, the predominant protease in the hepatopancreas, released to the freshwater prawn abdomen, mainly contributes to muscle softening during storage. Since the hepatopancreas is the major source of proteases, beheading or cephalothorax removal could lower the proteases contaminated in crustacean meat. Thus, beheading could prevent the degradation of muscle proteins, improve the quality and extend the shelf life of shrimps [4]. When protein degradation occurs, the peptides produced can be the nutrients for microbial growth, and the proliferation of microorganisms could favor the spoilage of HMS [2].

Additionally, the use of non-thermal processing, e.g., pulsed electric field (PEF), cold plasma and high-pressure processing (HPP), could prevent the quality loss caused by proteolysis and microorganisms and also extended the shelf life of shrimp [5]. PEF works based on electroporation and disintegration by recurrent pulses to the subject located between two electrodes [6]. It is an eco-friendly and residue-free technology, in which the quality and nutrients can be maintained [7]. Moreover, PEF is one of the non-thermal food processing technologies that could be implemented for the pretreatment of the samples to create tiny holes in the hard shell and intact shrimp meat [8]. In addition, PEF could inhibit polyphenol oxidase (PPO) and inhibit the growth of microorganisms. Gilliland and Speck [9] inactivated trypsin and protease from *Bacillus subtilis* using an electric field of 31.5 kV/cm. PEF was also effective in inactivating the microorganisms, especially spoilage and pathogenic bacteria such as *Pseudomonas* spp. and *Escherichia coli*, respectively [7]. The inactivation of microbes can be achieved by electroporation caused by intermittent pulses of high voltage, causing the breakdown of cell membranes and cell death [10].

Chitooligosaccharide (COS) is a derivative obtained from the hydrolysis of chitosan. COS is a bioactive compound readily soluble in water and has excellent antioxidant and antimicrobial properties [11]. COS inhibits the oxidation of proteins, lipids and microbial growth in seafoods. COS has also exhibited inhibitory activity against polyphenol oxidase (PPO), a metalloprotease containing copper atoms at its active site, responsible for melanosis in crustaceans. Thus, a COS solution could be used for preserving HMS during chilled or iced storage.

PEF shows enzyme inhibition and microorganism inactivation, whereas COS has antimicrobial, antioxidant and enzyme inhibitory activities. Considering these advantages, HMS was subjected to pretreatment with the PEF to create tiny pores in the hard shells. As a consequence, COS solution could penetrate via the pores into the PEF-pretreated HMS. However, no information on using prior PEF combined with COS treatment for the prevention of proteolysis and the shelf life extension of HMS exists. Therefore, this study aimed to illuminate the influence of the combined process of prior PEF and COS treatment on the changes in the quality of HMS without and with beheading during storage in iced water.

## 2. Materials and Methods

### 2.1. Chemicals

All chemicals of analytical grade were purchased from Sigma Aldrich (St. Louis, MO, USA). All media for microbial analyses were procured from Hi-Media Laboratories (Mumbai, India).

### 2.2. Collection and Preparation of the Samples

#### 2.2.1. Collection of Samples

Live HMS from Satun dockyard were caught during September and October 2023. The samples with a length of 13–18 cm were packed in a polystyrene box with cold water containing crushed ice by the vendors. The samples were brought to the lab within 2 h. Upon arrival, the samples were washed in cold water and stored in insulated boxes filled with crushed ice within 1–2 h.

#### 2.2.2. Preparation of Sample

The washed samples were categorized into two groups: (i) whole HMS, and (ii) beheaded HMS, where the cephalothorax was removed from the abdomen.

### 2.3. Pretreatment of Shrimp with PEF

The PEF pretreatment was performed following the method tailored by Sheikh and Benjakul [8], with slight modification. The cleaned whole and beheaded HMS samples were placed in a PEF chamber with a high-voltage power supply (PEF LAB-400w, Febix International Inc., Chiang Mai, Thailand). The system consisted of a chamber of 5 L capacity equipped with a stainless steel electrode (35 cm × 20 cm), an aluminum-framed plexiglass cabinet, a digital storage oscilloscope (DSO) (UTD2052CEX, UNI-T^®^, Dongguan city, Guangdong, China) and a UT-P03 oscilloscope probe (600 MHz, 10×) connected to a data logger. The samples were placed horizontally in a treatment chamber attached to plastic frames at the bottom, thus avoiding direct contact with the bottom electrode. HMS were then closed with the top electrode. The chamber was filled with sterile cold water (3 L) with crushed ice to maintain the temperature below 1 °C and to ensure that the electrode was fully covered. A uniform electrode gap of 3.0 cm was maintained. PEF (15 kV/cm, 800 pulses) with specific energy of 697 kJ/kg was applied.

### 2.4. Preparation of COS and COS Soaking of PEF-Treated Shrimp

For the synthesis of COS, the redox pair hydrolysis approach using ascorbic acid/H_2_O_2_ was used, as tailored by Mittal et al. [11]. COS had an average molecular weight (MW) of 0.7 kDa, a degree of polymerization of 2–8 and a degree of deacetylation of 91%.

COS solutions were prepared by dissolving COS at 1% and 2% (*w*/*v*) in 1 mL of 0.05% acetic acid for complete solubilization of COS and made up to 200 mL with distilled water. For 0% COS solution, the preparation was made in a similar manner, except COS was excluded. The solutions (pH = 6.8) were stored at 4 °C. The PEF-treated samples, both whole and beheaded samples, were soaked in COS solution for 20 min, drained on screen for 5 min, and stored in a zip lock bag. Packed samples were immersed in iced water at 4 °C for 3 days. The whole shrimps with prior PEF and treated with COS at 0, 1 and 2% COS solutions were referred to as CON-W, 1% COS-W and 2% COS-W, respectively. Beheaded shrimp with prior PEF and soaked in COS solutions at 0, 1 and 2% were named CON-BH, 1% COS-BH and 2% COS-BH, respectively. The samples soaked in distilled water were used as control. Samples were taken every day for analyses.

### 2.5. Extraction and Activity Assay of Trypsin in Muscle of HMS

#### 2.5.1. Trypsin Extraction

Trypsin contaminated in HMS muscle in all treatments was extracted. HMS mince (2 g) was mixed with four volumes of cold 50 mM Tris-HCl buffer containing 10 mM CaCl_2_ (pH 8.0) and homogenized (11,000 rpm, 2 min). The homogenate was centrifuged for 10 min at 4 ± 1 °C at 1100× *g* (Beckman Coulter, Avanti J-E Centrifuge, Fullerton, CA, USA). The supernatant was tested for trypsin activity.

#### 2.5.2. Trypsin Activity

Trypsin activity was determined using BAPNA as substrate at pH 8.0 and 60 °C for 20 min. After termination using 30% acetic acid, the amount of *ρ*-nitroaniline released during the reaction was measured by reading the absorbance at 410 nm using a spectrophotometer (Shimadzu, Kyoto, Japan). Trypsin activity was computed using the equation below. $$\mathrm{Trypsin}\;\mathrm{activity}=\frac{\left(A-\mathrm{Ao}\right)\times \mathrm{mixture}\;\mathrm{volume}\;\left(\mathrm{mL}\right)\times 1000}{8800\times \mathrm{reaction}\;\mathrm{time}\;(\min)\times 0.2}$$ where A and A_0_ are the absorbance of sample and blank, respectively; 8800 cm^−1^M^−1^ is *ρ*-nitroaniline coefficient. One unit of activity was defined as the amount of trypsin that released 1 nmol of *ρ*-nitroaniline per min under specified conditions.

### 2.6. Quality Assessment of HMS Meat

#### 2.6.1. Color

Color of meat from whole and beheaded HMS treated under different conditions was recorded based on CIE lab values (*L**, *a** and *b**). Hunter Lab (Colorflex, Reston, VA, USA) was used for color measurement.

#### 2.6.2. Shear Force

Shear force of HMS meat was examined as detailed by Chanchi Prashanthkumar et al. [3]. Shear force of five segments of the abdomen was measured with the assistance of TA-XT_2_ texture analyzer (Stable Microsystems, Surrey, UK) assembled with a Warner-Bratzler blade. A crosshead speed (2 mm/s) and load cell (50 kg) were used. The blade was applied perpendicular to the axis of the muscle fibers, and the force required to cut muscle was measured. Force–distance curves were employed for the calculation of shear force (N).

#### 2.6.3. Protein Patterns

SDS-PAGE was used to determine the protein patterns of meat from whole and beheaded samples with various treatments. The minced meat was dissolved in hot 5% SDS (85 °C). The sample (30 mg protein, as determined by the Biuret method) was loaded, and proteins were separated through 4% stacking gel and 12% running gel. Protein bands were then stained using Coomassie Brilliant Blue R250, followed by destaining. The molecular weight (MW) of protein bands was calculated using the relative migration distance (Rf) compared to those of protein standards.

#### 2.6.4. Trichloroacetic Acid (TCA) Soluble Peptide Content (TCA-SPC)

TCA-SPC of meat from whole and beheaded samples having various treatments was examined. Minced meat (3 g) was homogenized (11,000 rpm, 1 min) with 27 mL of a cold 5% TCA solution. Homogenates were centrifuged (5000× *g*, 4 °C, 20 min). The Lowry method was used to examine soluble peptides in the supernatant [12]. The content was expressed as µmol tyrosine equivalent/g mince.

#### 2.6.5. Histological Images

The sample (5th abdomen segment) was cut in transversal section and was stored in Davidson’s fixative solution and subsequently subjected to histological analysis using the method detailed by Hiransuchalert et al. [13] and fixed in Paraplast^TM^ (Histosec^®^, Merck, Darmstadt, Hesse, Germany). The staining and histological analysis were accomplished. The paraffin tissues were dissected into sections of 4 µm by a microtome and then stained using hematoxylin and eosin. The tissue was observed underneath a VDO capture digital camera (ECLIPSE Ni-U) 180 (Nikon, Tokyo, Japan).

Mean grey index (MI), with a pre-set scale bar at 400× magnification, was used to estimate the density or integrity of the tissue. Individual pictures (*n* = 10) were analyzed using ImageJ software 1.53 (National Institutes of Health, Bethesda, MD, USA) [14]. The data were computed and expressed as color intensity.

Anatomical observation of treated whole and beheaded HMS samples at day 0 and 3 was performed to examine the softening of digestive tract by lateral dissection.

#### 2.6.6. Fourier Transform Infrared (FTIR) Spectra

FTIR spectra of the meat of all the samples were obtained using a method tailored by Li et al. [15] with slight modifications. The 3rd and 4th segments of the HMS samples were freeze-dried with the aid of Scanvac Model Coolsafe 55 freeze dryer (Coolsafe, Lynge, Denmark). Dried samples were placed on the diamond crystal ATR at room temperature. The FTIR spectra of HMS samples were recorded with a Bruker Vertex 70 FTIR spectrometer (Bruker Co., Ettlingen, Germany) with a wavelength from 400 to 4000 cm^−1^, a scanning time of 32 times, and a resolution of 4 cm^−1^. The air background spectrum was recorded at 25 °C.

#### 2.6.7. Microbial Load

Meat of HMS sample was collected under sterile conditions and subjected to microbial analysis using the spread plate method proposed by Palamae et al. [16], with slight modifications. Five grams of HMS meat was diluted with 45 mL of 0.85% sterile saline solution and mixed for 1 min in the Stomacher blender (Stomacher M400, Seward Ltd., Worthington, UK). The homogenate obtained was subjected to serial dilutions and spread-plated (100 μL) in triplicates on plate count agar (PCA) (Code: M091, HiMedia Laboratories LLC, Mumbai, India) and incubated at 37 ± 1 °C for 48 h for total viable count (TVC). Psychrophilic bacteria count (PBC) was determined using PCA and incubated at 4 °C for 10 days. *Pseudomonas* spp., *Enterobacteriaceae* and *Shewanella* spp. were also determined by spread plate method using 100 μL of diluted samples on selective media. To enumerate *Pseudomonas* spp., the diluted homogenate was spread on *Pseudomonas* isolation agar (Code: M120, HiMedia Laboratories LLC, Mumbai, India) and incubated at 25 °C for 48 h. *Enterobacteriaceae* count was determined using Eosin methylene blue EMB agar and incubated at 37 °C for 48 h (Code: M317, HiMedia Laboratories LLC, Mumbai, India). For *Shewanella* spp. enumeration, the samples were plated on thiosulfate citrate bile salt agar (TCBS) (Code: M870S HiMedia Laboratories LLC, Mumbai, India) and incubated at 37 °C for 48 h. The count was reported as log CFU/g.

### 2.7. Statistical Analysis

Completely randomized design (CRD) was utilized, and the analyses were run in triplicates. Three different lots of HMS were used. Data were subjected to a one-way Analysis of Variance (ANOVA). The means were compared by Duncan’s multiple range test using an SPSS package (SPSS 23.0 for Windows SPSS Inc., Chicago, IL, USA).

## 3. Results and Discussion

### 3.1. Effect of Prior PEF and COS Treatment on Trypsin Activity in Meat of HMS

The trypsin activity significantly increased (*p* < 0.05) from day 0 (0.03 U/g) to day 3 (0.37 U/g) of storage in the CON-W samples. Similarly, the trypsin activity of 1% COS-W and 2% COS-W trypsin increased from 0 U/g to 0.35 and 0.30 U/g, respectively, from day 0 to day 3. All the treated samples with 1% and 2% COS showed lower trypsin activity (*p* < 0.05). A drastic 3-fold increase in trypsin activity of the HMS meat was found on day 3. This signified that the prior PEF and COS treatment could retard the increase in trypsin activity until day 2 for the whole HMS. However, Chandhi Prashanthkumar et al. [3] reported the rapid augmentation of the trypsin activity on day 1 of storage and remained constant till day 3. This was probably due to differences in the season, habitat, feeds, etc., which could determine the trypsin activity in the different HMS samples. Trypsin mainly caused the rapid protein degradation of the HMS muscle. Thus, the samples treated with COS displayed an inhibition of trypsin to a high degree till day 2 but still had trypsin activity at some level on day 3. The 2% COS-W sample showed lower trypsin activity than the 1% COS-W sample on day 3. For CON-BH, the trypsin activity was augmented significantly from day 1 (0 U/g) to day 3 (0.20 U/g) (*p* < 0.05). It was noteworthy that the digestive fluid was drained after decapitation. However, a small amount of trypsin still remained inside the cavity of the abdomen, where the digestive tract of the HMS was located. Thus, trypsin activity was still detected in the behead samples. Similarly, the 1% COS-BH and 2% COS-BH samples had trypsin activities of 0.14 U/g and 0.10 U/g, respectively, on day 0 to day 3 of storage. Among all the samples, the 2% COS-BH sample showed the lowest trypsin activity (*p* < 0.05) at all storage times, as illustrated in Figure 1.

In general, the meat of the whole and beheaded HMS samples with prior PEF, followed by soaking in COS solution, especially at 2%, possessed lowered trypsin activities. Between the whole and beheaded samples, the trypsin activities were higher in the former, where the digestive enzymes were entrapped inside the sample abdomen. When beheading was performed, the digestive liquid in the cavity of the abdomen was drained. However, no difference in the trypsin activity was found between the beheaded samples without and with prior PEF treatment. In addition, the PEF pretreatment combined with the COS soaking could inactivate the trypsin to some extent. A high-intensity electric field of 15 kV/cm emitting pulses could partially inhibit trypsin contaminated in muscle, as shown in Figure 1. 

Li et al. [17] documented that the high intensity of PEF inhibited trypsin. Similarly, Gilliland and Speck [9] also reported the complete inhibition of *Bacillus subtilis*’s protease and bovine trypsin with a high electric field of 31 kV/cm. PEF inhibited several enzymes such as peroxidase and papain, as reported by Van Loey et al. [18], de Lourdes Meza-Jiménez et al. [19] and Yeom et al. [20]. During the soaking of the PEF-treated sample, particularly the beheaded one, the COS solution could pass through the cavity (digestive tract), thus inhibiting trypsin more effectively. The combined prior PEF and COS treatment could therefore lower the trypsin activities, especially in the beheaded samples.

### 3.2. Effect of Prior PEF and COS Treatment on Quality of HMS Meat

#### 3.2.1. Color

Color is one of the major visual attractions to consumers, indicating the freshness and quality of the meat. The color of HMS meat reported as *L**, *a** and *b** values is shown in Table 1. The glossy grey meat of the HMS consisting of pigmented stripes on the ventral side showed an increase in the *L** value from day 0 to day 2–3, signifying an increase in lightness, more likely due to protein aggregation or denaturation. These results aligned with those of Temdee et al. [2] and Chanchi Prashanthkumar et al. [3]. It was noteworthy that the *L** values of the beheaded samples were higher than those of the control (without COS treatment). This might be due to the higher cross-linking of the proteins induced by COS, especially at 2%, via their amino groups, leading to the larger aggregate with a light scattering effect, as shown by the augmented lightness. The *a** values decreased up to day 2, followed by the increase at day 3 (*p* < 0.05). The highest *a** value at day 3 was found in the whole samples, regardless of the COS concentrations used. When the muscle underwent degradation and deterioration, carotenoproteins were also cleaved via proteolysis, thus liberating free astaxanthin. Astaxanthin contributes predominantly to the red or orange color of the shrimp. The *b** value was augmented as the storage time increased (*p* < 0.05). The concentrations of COS had no marked impact on the *b** value up to day 2. Nonetheless, there was an increase in the *b** value in the whole HMS meat on day 3, indicating the intensification of the yellowness of the HMS meat. More degraded proteins associated with a higher amount of free amino groups might undergo the Maillard reaction to a higher degree in comparison with the beheaded samples, in which lower trypsin activity was found.

#### 3.2.2. Shear Force

The change in shear force of the meat from the HMS, both the whole and beheaded samples, pretreated with PEF and soaked in COS solution at different levels from day 0 to day 3 of storage, is depicted in Figure 2. At day 0, for the whole sample, the COS treatment increased the shear force, compared to the CON-W samples. Nonetheless, the beheaded samples treated without and with 1 and 2% COS showed a similar shear force (*p* > 0.05). The increase in the strength of the meat was plausibly due to the absorption of the COS by the HMS meat. PEF could create the pores on the HMS’s hard shell, thereby allowing the COS solution to penetrate into the meat of the HMS. However, the CON-W sill had trypsin at some levels (Figure 3). As a result, the degradation of protein still occurred to some degree, as displayed by the slightly lower shear force. On day 1, a similar texture trend was observed where the whole sample showed a lower shear force, while the beheaded sample treated with 2% COS had the highest shear force (*p* < 0.05). Trypsin entrapped in the cavity of the HMS abdomen mainly caused the proteolysis and weakening of muscle in the whole HMS. However, the PEF-COS-treated whole sample showed a higher shear force than the CON-W sample. On day 2 and 3, the control samples, both the whole and beheaded samples, showed the least shear force, compared to the COS-treated samples. As mentioned previously, the COS absorption could take place and interact with the muscle proteins in the HMS meat. Moreover, the PEF-treated beheaded samples could absorb more COS, compared to the whole sample. Overall, the 2% COS-BH sample had the maximum shear force during the storage for 3 days, confirming the highest efficacy in retaining the texture of the HMS meat.

#### 3.2.3. Protein Pattern

The protein patterns of all the PEF-treated HMS samples, both whole and beheaded, soaked in 0, 1 and 2% COS solutions during 3 days of storage are depicted in Figure 3A and Figure 3B, respectively. For the whole HMS sample, myosin heavy chain (MHC) in CON-W totally disappeared, and actin (AC) was also degraded. The high degradation of MHC might be due to the stabilization of trypsin caused by PEF. Li et al. [17] reported that trypsin stabilized by the PEF with the high-intensity electric field resulted in the degradation of MHC. However, the HMS samples pretreated with PEF and soaked in 1 and 2% COS had retained MHC, AC and TNT bands. This signified that the combined treatment of PEF and COS could prevent the degradation of muscle proteins. The result was consistent with trypsin activity, which became lower after such treatments. However, on day 3, drastic decreases in the band intensity of MHC, AC and TNT were found, except 2% COS-W, which still had a high band intensity of MHC, AC and partial TNT. The results indicated the efficiency in the prevention of the protein degradation of 2% COS treatment of the PEF-pretreated HMS.

The protein patterns of the beheaded HMS sample pretreated with PEF and soaked in 0, 1 and 2% COS solutions were similar at day 0. Moreover, MHC, AC, TNT and other notable protein bands were retained, irrespective of the COS treatments. This indicated that the process of beheading and the draining of the digestive liquid entrapped in the cavity of the HMS abdomen was crucial. Thus, beheading could be an adequate process to maintain structural proteins for a short period. On day 3, MHC, AC, TNT and other proteins were drastically degraded in the CON-BH sample. Most of the proteins were retained in the PEF-pretreated samples and soaked in 2% COS. Therefore, the 2% COS treatment had a maximum efficacy in preventing the proteolysis of the beheaded sample. The results aligned with trypsin activity and texture, where a high shear force was still found after day 3. Additionally, COS could inactivate trypsin localized in the cavity with ease in the beheaded samples. Overall, the combination of the PEF pretreatment and 2% COS treatment of the beheaded HMS effectively retained muscle proteins after storage in iced water for 3 days.

#### 3.2.4. TCA-SPC

The protein degradation in all the HMS samples, expressed as the TCA-SPC, is presented in Figure 4. At day 0, the TCA-SPC varied from 0.08 to 0.45 µmol tyrosine/g in all the samples; both the whole and beheaded samples. The CON-W sample showed the highest TCA-SPC, which implied the action of PEF in the stabilization of trypsin in the digestive tract. This coincided with the marked degradation of most muscle proteins (Figure 3). The COS-soaked whole samples, including the 1% COS-W and 2% COS-W samples, showed a lower TCA-SPC, by 44.04% and 48.69%, respectively. A reduction in degradation by 82.72% and 84.35%, respectively, were found in the beheaded samples. On day 1, the whole samples showed a higher TCA-SPC than the beheaded samples, regardless of COS treatment. The CON-W sample had the highest TCA-SPC. The descending order of the TCA-SPC was found as follows: 1% COS-W, 2% COS-W, CON-BH, 1% COS-BH, and 2% COS-BH, in which the TCA-SPC was less than that of the CON-W sample by 17.19%, 34.79%, 80.66%, 83.83% and 86.72%, respectively. Similarly, a trend was found on day 2 and day 3. The 2% COS-BH sample had the lowest TCA-SPC, which was lower than that found in the CON-W sample by 73.77% and 74.63% on day 2 and 3, respectively. In general, the increase in the TCA-SPC was noted from day 0 to Day 3. The TCA-SPC was increased by 3–4 folds at day 3. All the results aligned with the increased trypsin activity, upsurged degradation of muscle proteins and lower shear force for the CON-W sample as the storage time upsurged. However, the 2% COS-BH sample had much lower changes. The TCA-SPC found in all samples on day 0 was probably due to the partial degradation of muscles caused by high-intensity PEF. In addition, PEF could stabilize the trypsin, which was responsible for the degradation of MHC. Thus, this phenomenon favored the hydrolysis of MHC and other proteins. Chanchi Prashanthkumar et al. [3] reported that endogenous enzymes were involved in the proteolysis or hydrolysis of muscle proteins. Notably, the 2% COS-BH sample displayed resistance to hydrolysis by endogenous enzymes, especially trypsin.

#### 3.2.5. Histological Images

The histological or microstructural changes in muscle tissue correspond to the texture and quality changes in meat [21]. Overall, three distinct muscle tissue changes, including autolysis, causing loss of muscle striation and architecture, fiber splitting or gapping of muscle fibers and disappearance of muscle fibers, were observed. These changes were varied with samples as influenced by different treatments (Figure 5A). The fresh muscles exhibited a significantly tight structure with no discernible histological changes. At day 0, the 1% COS-W and 2% COS-W samples showed the order alignment of fibers, while the CON-W sample had the disintegrated arrangements of fibers or disappearance of fibers. This was related to the complete disintegration of MHC (Figure 5A(I)). After 3 days of storage, the gapping was more evident for those treated with COS at both levels. The CON-W sample underwent more disruption of the muscle fibers. On the other hand, the BH sample treated with 2% COS (2% COS-BH) still possessed a more ordered fiber structure than the others. This was confirmed by the maintenance of the MHC and actin of this sample as indicated by the SDS-PAGE protein pattern (Figure 3). Consequently, this tight and dense muscle structure rendered a very high mean grey intensity (MI), contributing to a darker appearance (Figure 5B). In general, the muscles of the 2% COS-BH sample on day 0 exhibited the highest MI values. Similar results were found for the 1% COS-BH and 2% COS-BH samples, which showed a higher MI value than the CON-BH sample. Those samples displayed similar trends, compared to those found on day 3. Overall, the MI value was higher in the BH samples than the whole sample. This suggested that the decapitation provided tissue stability, as witnessed by the higher MI values.

Coincidently, the anatomical observation supported the above histological images as illustrated in Figure 5C. The fresh sample on day 0 had firm muscles, and the digestive tract was found intact in the CON-W, 1% COS-W and 2% COS-W samples. Nonetheless, the CON-BH, 1% COS-BH, and 2% COS-BH samples had firmer muscles and well-organized empty digestive cavities, compared to the whole samples. The whole samples rich in trypsin underwent degradation to a high extent, in which the muscle was destructed and softened. The PEF pretreatment electroporated the hard shell of the HMS, and a larger amount of COS was able to penetrate into the muscle, especially when a higher concentration of COS was used. On day 3, the whole sample had a mushy muscle texture. The CON-W sample was more mushy or pasty, compared to the other samples, including the beheaded ones. Among all the samples, the 2% COS-BH sample had the least mushiness. Thus, it provided a prominent significance of the COS treatment and beheading for preventing the muscle softening of the HMS. The anatomical data coincided with the shear force and histological images.

#### 3.2.6. FTIR Spectra

The FTIR spectra of all the freeze-dried samples with the PEF pretreatment and soaked in 0, 1 and 2% COS solutions on days 0 and 3 are illustrated in Figure 6A,B. All the samples’ FTIR spectra exhibited amide regions, which included amide I (1600–1700 cm^−1^), amide II (1500–1600 cm^−1^), amide-III (1320–1220 cm^−1^), amide A (3290–3400 cm^−1^) and amide B (2926 cm^−1^) [21,22,23,24]. Amide I indicates the vibrations of the carbonyl (C=O) stretch. Jose et al. [22] stated that the structure of the protein conformation depends on the hydrogen bond absorption peak in the amide I region. Amide II indicates amine groups, the N-H and also C-N stretch. Moreover, amide II has symmetry vibrations for the C-N-C=O absorption peak at 1499 cm^−1^. Amide III is associated with amine groups similar to amide II and the helical structure of the protein [22]. The amide A region of the NH stretching exclusively localized on the protein could be detected at a peak between 3290–3400 cm^−1^ [23]. Similarly, amide B, representing the C-H stretch for the mantis shrimp, could be found at 2916–2940 cm^−1^ [13]. In general, no differences in the amide I, II and III regions were found among any of the samples. When the HMS samples were pretreated with PEF and soaked in COS solutions before storage for 3 days, the intensity of the absorption peak at 2362 cm^−1^ was augmented steadily. This increase indicated the upsurge in the bending vibration of the N-H bond. These results were consistent with a higher degradation of proteins, in which free amino groups could be more liberated. Also, the amplitudes of amide A increased after 3 days of storage. This confirmed that more free amino acid was present in the sample.

The secondary structure analysis based on the amide I band (1600–1700 cm^−1^) was carried out. The deconvolution of the amide I band provided data on the α-Helix, β-turns, random coiling and β-sheets with the center at 1650–1660 cm^−1^, 1660–1680 cm^−1^, 1640–1650 cm^−1^, 1610–1640 and 1600–1700 cm^−1^, respectively [25,26,27]. The prior PEF and COS treatment had a minimal impact on the protein structures, regardless of the storage time (Figure 6B). In general, the percentage of β-sheets, α-helix, β-turns and random coil of all the samples were similar. On the other hand, there was no difference in all the structures, irrespective of the treatments of COS, storage time and beheading. Among all the structures, β-sheets constituted the dominant structure in the HMS muscle protein, constituting around 44–45%. The second dominant structure was β-turns, followed by random coil and α-helix, respectively. It was noteworthy that all the samples were stored in iced water with a temperature of 3–4 °C. This temperature was much lower than the denaturation temperature of MHC and actin, the major proteins in the HMS muscle. Although those proteins were hydrolyzed by trypsin, the confirmation was still retained since no energy was applied to destabilize the intermolecular and intramolecular bonds of the protein structure. As a consequence, no change in the secondary structure was noticeable. Thus, protein degradation was not directly related to conformational changes known as denaturation. This was because the samples were kept at temperatures below the denaturation temperature of the muscle proteins.

#### 3.2.7. Microbial Load

Various bacteria play a key role in the spoilage of seafoods and their products. Microbial enzymes cause the proteolysis of HMS during storage. TVC, PBC, *Pseudomonas* spp., *Enterobacteriaceae* and *Shewanella* spp. counts of all the samples are depicted in Figure 7A–E, respectively. On day 0, the TVC of the HMS sample pretreated with PEF and soaked in 0, 1 and 2% COS solutions were in the range of 3.35–3.93 log CFU/g sample. The PBC was almost equivalent to the TVC (3.54–3.74 log CFU/g). The *Pseudomonas* spp. count (3.00–3.19 log CFU/g), *Enterobacteriaceae* count (3.00–3.50 log CFU/g), and *Shewanella* spp. count (3.00–3.15 log CFU/g) were also detected at day 0. In general, there was no difference in the load of all the microorganisms tested between all the samples, irrespective of beheading or the COS concentrations used at day 0. On day 1, the CON-W and CON-BH samples had the highest TVC (4.58 and 4.89 log CFU/g, respectively) and PBC (4.10 and 4.18 log CFU/g, respectively). However, the other microorganisms tested were not affected by the COS treatment or beheading. As the storage time increased, the TVC exceeded the limit of (10^6^ CFU/g) after day 1. The continuous increase in the microbial load with the augmenting of the storage time was noted. From day 2 to day 3, similar loads were observed for the TVC, PBC, *Pseudomonas* spp., and *Shewanella* spp. counts, except 2%COS-BH sample, which had TVC lower than the limit at day 2. The increase in all the microorganisms tested might be attributed to the increase in protein degradation, which provided free amino acids or small peptides as nutrients for microorganisms. This coincided with the increased TCA soluble peptide content and the disappearance of most of the muscle proteins after storage for 3 days. Peptides were formed, caused by the proteolysis of the HMS muscles and were mainly mediated by endogenous enzymes, particularly trypsin [3]. It was noted that COS with a higher concentration (2%) yielded HMS meat with a lower load of psychrophilic bacteria and *Enterobacteriaceae* in the beheaded samples than the others at day 0 of storage. PEF has also been reported to inactivate and kill bacteria by electroporation [28]. COS is known for its impeccable antimicrobial properties against spoilage bacteria such as *P. aeruginosa* and pathogenic bacteria such as *Listeria monocytogenes* and *Vibrio parahaemolyticus* [11]. For the beheaded samples, the COS solution was able to reach the cavity, where some microbes were present. COS could alter cell walls and membranes [11]. Additionally, COS might enter the microbial cell, resulting in changes in different biomolecules. Ultimately, this phenomenon either stopped the microbial growth or resulted in cell death [29]. Additionally, the COS-mediated chelation of vital nutrients for microbial growth may cause the inactivation of some microorganisms [11]. However, the sensitivity of the different microorganisms toward COS was varied in the present study. Thus, PEF in combination with a higher concentration of COS (2%) showed the potential to inhibit the growth of some spoilage bacteria to a certain degree.

## 4. Conclusions

PEF pretreatment in combination with 2% COS of beheaded HMS could lower trypsin activity in HMS meat and delay the autolysis process. The texture of the HMS was maintained till day 3 of storage in iced water, as evidenced by the retained MHC, AC and TNT in the meat with coincidentally lowered TCA soluble peptides and retained muscle fiber integrity. There was no difference in the secondary structure of the muscle protein among all the treatments. The decapitated samples treated with 2% COS solution showed the lowest microbial count. Nevertheless, the TVC of all the samples exceeded the microbial limit after 2 days. Thus, the HMS samples, especially beheaded ones, treated with PEF and COS at higher concentrations had a lowered softening, a major drawback of HMS during postmortem storage. Nevertheless, PEF treatment might bring about the higher cost of operation, while beheading might make the meat more prone to microbial contamination. To enhance the penetration of COS into the HMS meat or cavity rich in digestive enzymes, vacuum impregnation could be properly applied. As a consequence, the quality of the HMS can be maintained during post-harvest storage more effectively.

## Figures and Tables

**Figure 1 foods-13-00028-f001:**
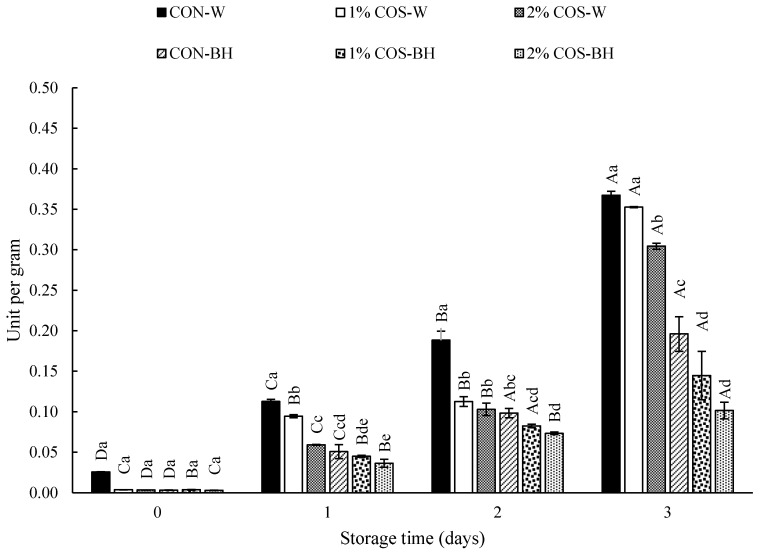
Effects of prior pulsed electric field and chitooligosaccharide at different concentrations on trypsin activity of meat from Harpiosquillid mantis shrimp (*Harpiosquilla raphidea*) during storage in iced water. Bars represent the standard deviation (*n* = 3). Different lowercase letters within the same storage times denote significant differences (*p* < 0.05). Different uppercase letters within the same sample denote significant differences (*p* < 0.05).

**Figure 2 foods-13-00028-f002:**
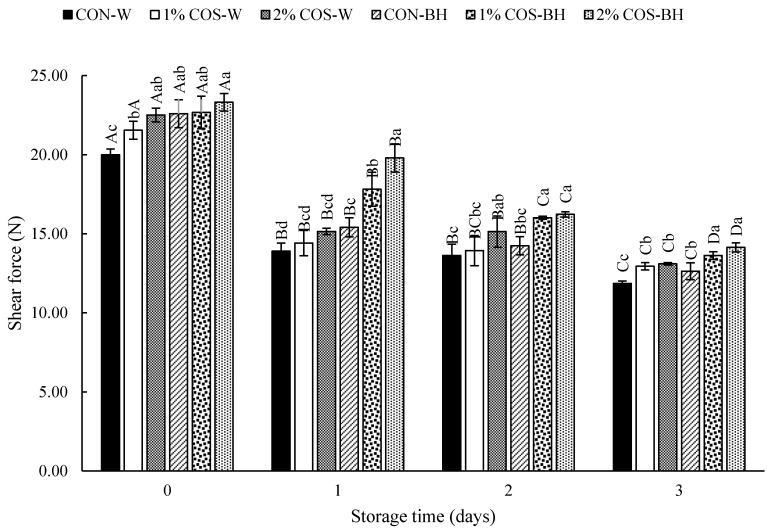
Effects of prior pulsed electric field and chitooligosaccharide at different concentrations on shear force of meat from Harpiosquillid mantis shrimp (*Harpiosquilla raphidea*) during storage in iced water. Bars represent the standard deviation (*n* = 3). Different lowercase letters within the same storage times denote significant differences (*p* < 0.05). Different uppercase letters within the same sample denote significant difference (*p* < 0.05).

**Figure 3 foods-13-00028-f003:**
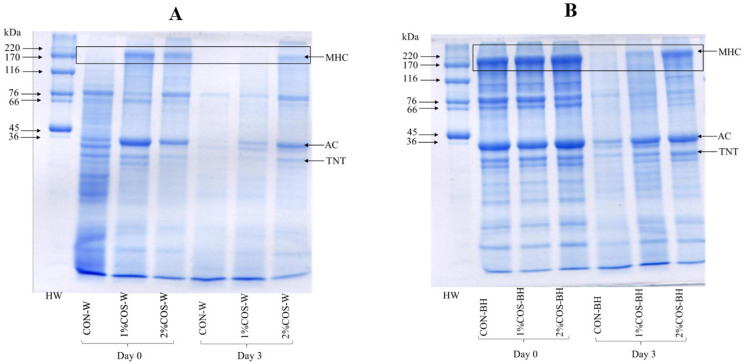
Effects of prior pulsed electric field and chitooligosaccharide at different concentrations on protein pattern of meat from whole (**A**) and beheaded (**B**) Harpiosquillid mantis shrimp (*Harpiosquilla raphidea*) at day 0 and day 3 of storage in iced water. MHC: Myosin Heavy Chain; AC: Actin, TNT: Troponin T.

**Figure 4 foods-13-00028-f004:**
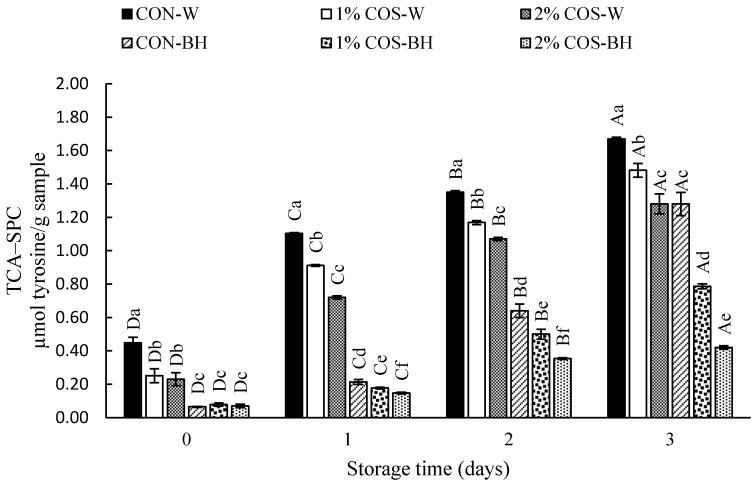
Effects of prior pulsed electric field and chitooligosaccharide at different concentrations on TCA soluble peptide content of meat from Harpiosquillid mantis shrimp (*Harpiosquilla raphidea*) during storage in iced water. Bars represent the standard deviation (*n* = 3). Different lowercase letters within the same storage times denote significant differences (*p* < 0.05). Different uppercase letters within the same sample denote significant differences (*p* < 0.05).

**Figure 5 foods-13-00028-f005:**
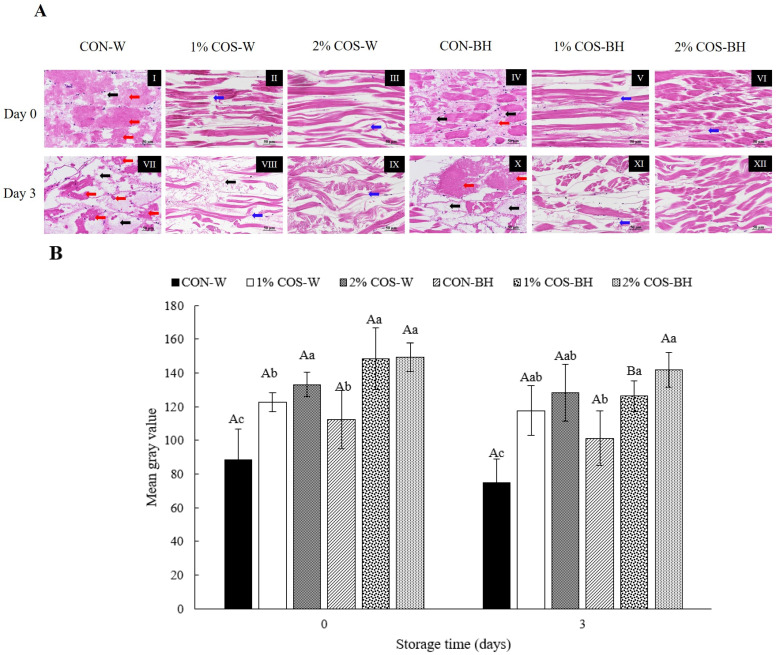
Effects of prior pulsed electric field and chitooligosaccharide at different concentrations on histomorphometric images (**A**), mean grey index value (**B**) and appearance (**C**) of meat from Harpiosquillid mantis shrimp (*Harpiosquilla raphidea*) at day 0 and day 3 of storage in iced water. The illustration depicts the fiber splitting/gapping (indicated by the blue arrow), autolysis indicating loss of muscle striation and architecture (indicated by the red arrow), and disappearance of muscle fibers (indicated by the black arrow). For muscle fiber orientation (Figure 5A), longitudinal (I–III, V, VIII, IX) and oblique (IV, VI, VII, X–XII) orientation were illustrated. Magnification: 40× Bars represent the standard deviation (*n* = 3). Different lowercase letters within the same storage times denote significant differences (*p* < 0.05). Different uppercase letters within the same sample denote significant differences (*p* < 0.05).

**Figure 6 foods-13-00028-f006:**
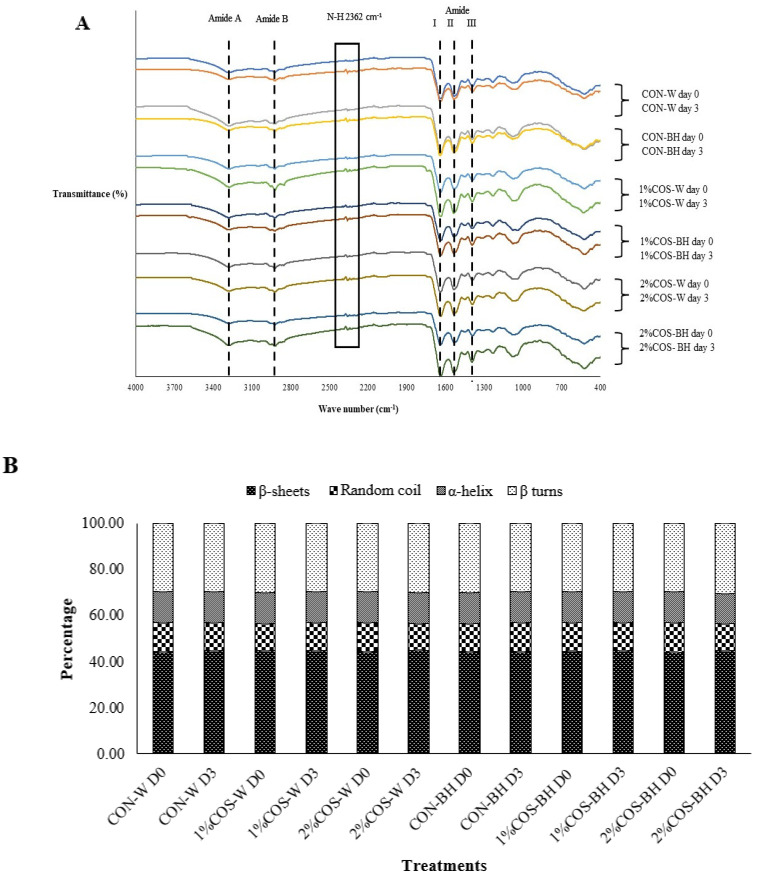
Effects of prior pulsed electric field and chitooligosaccharide at different concentrations on FTIR spectra (**A**) and secondary structure representing the β−sheets, random coil, α−helix and β−turns (**B**) of meat from Harpiosquillid mantis shrimp (*Harpiosquilla raphidea*) at day 0 and 3 of storage in iced water. D0 and D3 represent Day 0 and Day 3, respectively.

**Figure 7 foods-13-00028-f007:**
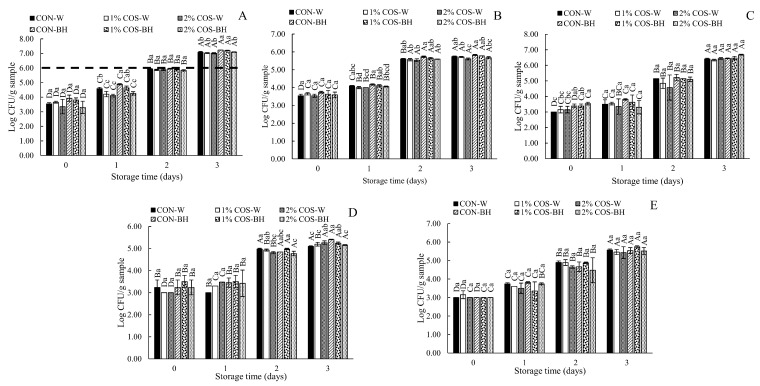
Effects of prior pulsed electric field and chitooligosaccharide at different concentrations on total count (**A**), psychrophilic bacteria count (**B**), *Pseudomonas* spp. count (**C**), *Enterobacteriaceae* count (**D**) and *Shewanella* spp. count (**E**) of meat from Harpiosquillid mantis shrimp (*Harpiosquilla raphidea*) during storage in iced water. Bars represent the standard deviation (*n* = 3). Different lowercase letters within the same storage times denote significant differences (*p* < 0.05). Different uppercase letters within the same sample denote significant differences (*p* < 0.05). The dashed line represents the safety limit of 6 log_10_ CFU/g.

**Table 1 foods-13-00028-t001:** Effects of prior pulsed electric field and chitooligosaccharide at different concentrations on color of meat from Harpiosquillid mantis shrimp (*Harpiosquilla raphidea*) during storage in iced water.

Storage Time (Days)	Treatments	Color		
		*L**	*a**	*b**
0	CON-W	49.40 ± 0.46 ^Bb^	−0.41 ± 0.03 ^Bab^	3.20 ± 0.07 ^Da^
	1% COS-W	50.90 ± 0.39 ^Ca^	−0.43 ± 0.02 ^Bab^	3.28 ± 0.12 ^Da^
	2% COS-W	47.41 ± 0.59 ^Cc^	−0.45 ± 0.04 ^Aabc^	3.30 ± 0.05 ^Da^
	CON-BH	46.6 ± 0.16 ^Cc^	−0.40 ± 0.01 ^Aa^	3.17 ± 0.06 ^Ca^
	1% COS-BH	44.41 ± 0.43 ^Cd^	−0.49 ± 0.04 ^Ac^	3.26 ± 0.02 ^Ca^
	2% COS-BH	41.85 ± 1.05 ^Be^	−0.46 ± 0.02 ^Abc^	3.30 ± 0.08 ^Da^
1	CON-W	51.79 ± 1.00 ^Ac^	−1.17 ± 0.04 ^Ca^	4.44 ± 0.18 ^Ca^
	1% COS-W	53.02 ± 0.39 ^Abc^	−1.11 ± 0.03 ^Ca^	4.59 ± 0.22 ^Ca^
	2% COS-W	54.18 ± 0.76 ^Ab^	−1.06 ± 0.05 ^Ba^	4.61 ± 0.13 ^Ca^
	CON-BH	57.19 ± 1.53 ^Aa^	−1.13 ± 0.13 ^Ba^	4.36 ± 0.17 ^Ba^
	1% COS-BH	57.84 ± 0.77 ^Aa^	−1.10 ± 0.06 ^Ba^	4.57 ± 0.34 ^Ba^
	2% COS-BH	57.53 ± 0.20 ^Aa^	−1.08 ± 0.03 ^Ba^	4.64 ± 0.08 ^Ca^
2	CON-W	51.45 ± 0.67 ^Ac^	−1.26 ± 0.13 ^Cb^	6.32 ± 0.38 ^Bb^
	1% COS-W	51.91 ± 0.51 ^ABbc^	−1.17 ± 0.05 ^Dab^	6.55 ± 0.12 ^Bab^
	2% COS-W	52.07 ± 0.82 ^Bbc^	−1.15 ± 0.09 ^Bab^	6.19 ± 0.19 ^Bb^
	CON-BH	53.16 ± 1.24 ^Bb^	−1.11 ± 0.05 ^Ba^	6.39 ± 0.34 ^Ab^
	1% COS-BH	56.64 ± 0.81 ^Aba^	−1.06 ± 0.04 ^Ba^	6.21 ± 0.06 ^Ab^
	2% COS-BH	57.04 ± 0.63 ^Aa^	−0.59 ± 0.02 ^Ba^	6.96 ± 0.32 ^Aa^
3	CON-W	50.97 ± 0.15 ^Ade^	−0.16 ± 0.02 ^Aa^	7.87 ± 0.11 ^Aa^
	1% COS-W	50.45 ± 0.36 ^BCe^	−0.22 ± 0.01 ^Aab^	7.42 ± 0.22 ^Aab^
	2% COS-W	51.82 ± 0.11 ^Bcd^	−0.38 ± 0.15 ^Aabc^	6.98 ± 0.09 ^Abc^
	CON-BH	51.92 ± 0.79 ^Bc^	−0.29 ± 0.06 ^Aab^	6.72 ± 0.56 ^Ac^
	1% COS-BH	55.66 ± 0.72 ^Bb^	−0.46 ± 0.24 ^Abc^	6.50 ± 0.27 ^Ac^
	2% COS-BH	56.45 ± 0.65 ^Aa^	−0.59 ± 0.21 ^Ac^	5.90 ± 0.42 ^Bd^

Values are given as mean  ±  SD (*n* = 5). Different lowercase superscripts in the same column within the same storage times denote significant differences between samples (*p* < 0.05). Different uppercase superscripts in the same column within the same sample denote significant differences (*p* < 0.05).

## Data Availability

The data are contained in the article.

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
