# Peer review of "Impact of Prior Pulsed Electric Field and Chitooligosaccharide Treatment on Trypsin Activity and Quality Changes in Whole and Beheaded Harpiosquillid Mantis Shrimp during Storage in Iced Water"

_foods, 2023, doi:10.3390/foods13010028_

Round 1
Reviewer 1 Report
Comments and Suggestions for Authors
I recommend a minor correction.
Authors need to introduce in all the analyses the Control without PEF and chitooligosaccharide treatment.
The presence of the control is necessary to justify the effect of the treatments and to provide an assembly view of the trypsin enzyme activity, samples colour, TCA-SPC content etc.
The control need to be provide for the 2 sample categories whole HMS and beheaded HMS.
Furthermore, transporting the samples within 2 hours on crushed ice from the time of capture to the laboratory allows cryophilic microorganisms to grow and may influence the results of the study. (Line 99-103)
Authors need to provide analyses for control sample for both categories whole HMS and be-headed HMS. Without the Control is very difficult to observe the conjugated effect of PEF and Chitooligosaccharide during shrimp storage in iced water.
Author Response
Reviewer 1
****Thank you so much for the invaluable comment and suggestion. All queries have been responded and the corrections have been made as highlighted in grey.
1. Authors need to introduce in all the analyses the Control without PEF and chitooligosaccharide treatment.
**** Thank you for your invaluable suggestion and your time in going through the manuscript. The authors would like to clarify that the control without PEF and chitooligosaccharides treatment was previously studied and reported in our published papers. Please see the following references.
References
Chanchi Prashanthkumar, M., Temdee, W., Suyapoh, W., Sornying, P., Palamae, S., Patil, U., Ma, L., & Benjakul, S. (2023). Effects of proteases on textural softening and changes in physical property of Harpiosquillid mantis shrimp (Harpiosquilla raphidea) during the chilled storage. International Journal of Food Science & Technology, 58, 6372-6385. https://doi.org/10.1111/ijfs.16745
Temdee, W., Singh, A., Zhang, B., & Benjakul, S. (2022). Effect of vacuum packaging on shelf-life extension of cooked and peeled harpiosquillid mantis shrimp (Harpiosquilla raphidea) during refrigerated storage. International Journal of Food Science & Technology, 57(7), 4451–4462. https://doi.org/10.1111/ijfs.15778
Hence, the authors avoid to report the same treatment repeatedly. Also, the current study aimed to elucidate the impact of beheading and COS treatment on the softening and quality changes of HMS with prior PEF treatment. HMS have the thick shell and we found in our preliminary study that PEF can generate the tiny pores, based on electroporation. These pores allow the COS solution to pass through the shell without peeling. Also, we observed that the whole mantis shrimp can undergo softening or drastic degradation of muscle proteins at day 0 without any treatment. For control beheaded sample, the sample showed no significant difference in trypsin activity between samples without and with PEF treatment. Those results have been reported in line 269-271. As a consequence, the control was not included in the present study.
2. The presence of the control is necessary to justify the effect of the treatments and to provide an assembly view of the trypsin enzyme activity, samples color, TCA-SPC content etc.
**** Thank you for your valuable suggestion. As mentioned earlier, the control sample without PEF and chitooligosaccharides treatment was completely analysed and reported in the authors' previous research (Chanchi Prashanthkumar et al., 2023),especially the trypsin activity, sample color and TCA-SPC content. Moreover, the results were compared with previous report in lines 250-251, 296-297 and 404-405, respectively. Furthermore, the aim of the research was to compare the notable changes in trypsin activities, color, texture, TCA soluble peptides, microbial load and secondary structure, which might be caused by synergistic effect between beheading and chitooligosaccharides soaking on the retardation of the aforementioned changes of PEF pretreated HMS.
3. The control need to be provide for the 2 sample categories whole HMS and beheaded HMS.
**** Thank you for your insightful comment suggestion. As mentioned above, the authors would like to compare the effect of beheading and COS treatment on the softening and quality changes of PEF pretreated HMS. Without PEF, the penetration through the whole mantis shrimp is negligible. In the present study, authors focused only the efficacy of both factors. Thus, the general controls (whole HMS and beheaded HMS) were excluded. In fact, those controls, especially the whole sample, were prone to softening, while no difference between beheaded HMS without and with prior PEF was found in all aspects (based on our preliminary study).
4. Furthermore, transporting the samples within 2 hours on crushed ice from the time of capture to the laboratory allows cryophilic microorganisms to grow and may influence the results of the study. (Line 99-103)
**** Thank you for the valuable comments. Due to the limitations of the remote raw material source at the research site, it was necessary to preserve the samples for at least 2 h during transportation. The author would like to clarify that the samples were brought to laboratory by embedding in crushed ice mixed with clean cold water for uniform distribution of temperature below 4°C. Due to the short time (2 h), a negligible or minimal growth of cryophilic microorganisms was presumed. In general, the microorganisms must take a long time like 7-10 days to adapt themselves to the new environment, especially cold atmosphere (4°C). Typically, a significant contributor to the deterioration in the quality of fresh aquatic foods is the presence of microflora, often attached to the organs of aquatic animals (Pan et al., 2019). The growth of psychrophilic or cryophilic bacteria is at optimum temperature at 15°C and minimal between 0-4°C (MOYER et al., 2017). In the present study, the temperature was maintained throughout the transportation for 2 h below 4°C. A previous study by Temdee et al. (2022) found that the psychrophilic bacteria of freshly harvested HMS were 2±0.8 log10 CFU/g after transporting the samples on crushed ice for 3 h. In the case of shrimp, Jeyasekaran et al. (2006) reported the psychrophilic bacteria of freshly harvested shrimp to be 5 log10 CFU/g. However, this psychrophilic population decreased by one log10 after 1 h of storage in a water-ice mixture with a 1:1 ratio (Jeyasekaran et al., 2006).
In the current study, when freshly harvested and transported HMS (day 0) were tested for psychrophilic bacteria, the results were 3.2 and 3.5 log10 CFU/g, respectively. Therefore, by transporting the samples within 2 h in water containing crushed ice from the time of capture to the laboratory, psychrophilic bacteria count was not significantly increased, compared to that found in freshly harvested HMS.
References
Jeyasekaran, G., Ganesan, P., Anandaraj, R., Jeya Shakila, R., & Sukumar, D. (2006). Quantitative and qualitative studies on the bacteriological quality of Indian white shrimp (Penaeus indicus) stored in dry ice. Food Microbiology, 23(6), 526–533. https://doi.org/https://doi.org/10.1016/j.fm.2005.09.009
Moyer, C., Collins, R., & Morita, R. (2017). En: Psychrophiles and psychrotrophs. Reference Module in Life Sciences. Chichester (Reino Unido). In: John Wiley & Sons.
Pan, C., Chen, S., Hao, S., & Yang, X. (2019). Effect of low-temperature preservation on quality changes in Pacific white shrimp, Litopenaeus vannamei: a review. Journal of the Science of Food and Agriculture, 99(14), 6121–6128. https://doi.org/https://doi.org/10.1002/jsfa.9905
5. Authors need to provide analyses for control sample for both categories whole HMS and be-headed HMS. Without the Control is very difficult to observe the conjugated effect of PEF and Chitooligosaccharide during shrimp storage in iced water.
**** Thank you for the invaluable comment on improving the manuscript. The authors found that the control sample without PEF and chitooligosaccharides were prone to complete softening (Chanchi Prashanthkumar et al., 2023). To further improve the quality maintenance or lowering the softening, the present research aims to compare changes in pretreated samples with PEF as influenced by beheading and COS treatment.
Reviewer 2 Report
Comments and Suggestions for Authors
1. The authors suggested that trypsin activity in different HMS samples was seasonally related (lines 250-251). Therefore, it is necessary to describe in detail the season and size of HMS (line 98).
2. Samples for FT-IR should be taken from all the meat after freeze-drying, not just from the 3rd and 4th segments.
3. Figure 1 showed the lack of standard deviation.
4. No reason was given for the increase in L* value after the samples (2% COS-BH) were stored for 3 days.
5. When determining the shear force, only 3 samples were determined (n=3), the number of samples was too small.
6. The histological images in Figure 5 presented as a cross-cut or a longitudinal cut, there was no uniformity. Besides, the different colored arrows in Figure 5 should indicate this.
7. 2% COS-BH Day 0 shows a surge in bending vibration of N-H bonds? Does this mean 2% COS-BH protein on day 0? Please explain.
8. It is recommended to put the microbiome load at the top to explain why it is only stored for 3 days (from 3.7 to 3.1).
9. PEF pretreatment combination containing 2%COS delayed the autolysis process of HMS meat. The effect of COS can be determined by comparing different concentrations, but there is no comparison of PEF pretreatment, so how do you confirm its effect?
Author Response
Reviewer 2
**** Thank you so much for the insight comments. All queries have been responded and the corrections have been done as highlighted in yellow.
1. The authors suggested that trypsin activity in different HMS samples was seasonally related (lines 250-251). Therefore, it is necessary to describe in detail the season and size of HMS (line 98).
**** Thank you for your invaluable suggestion. The season and size of HMS have been provided in line 99-100.
2. Samples for FT-IR should be taken from all the meat after freeze-drying, not just from the 3rd and 4th segments.
**** Thank you for the comment. Since the samples from different treatments were subjected to varying analyses. Therefore, to maintain the precise sampling process, the segment of abdomen was fixed for different assays or analyses. Since the 3rd and 4th segments are damaged severely due to the endogenous enzymes, mainly trypsin released from the digestive cavity into the muscle of HMS, both segments were selected for FT-IR analysis, especially for amide I peak analysis in order to study the changes in the secondary structure of muscle protein prone to cleavage by protease. Anatomical observation also confirmed the leakage of protease on the muscle deterioration surrounding the 3rd and 4th segments, particularly on day 3 of storage.
3. Figure 1 showed the lack of standard deviation.
**** As per the reviewer’s suggestion, the standard deviations (SD) were very low in values and the bars representing the SD were not clearly observed. Sorry for this.
4. No reason was given for the increase in L* value after the samples (2% COS-BH) were stored for 3 days.
**** Thank you for the invaluable suggestion. The possible reason for the increase in lightness of 2% COS-BH has been discussed as shown in line 298-302.
5. When determining the shear force, only 3 samples were determined (n=3), the number of samples was too small.
**** For analysis of shear force, 5 segments of each sample were subjected to shear force determination as mentioned in line number 169-170. Hence, the number of points checked for the texture was 15, which would be sufficient to fix the data on the shear force of the muscles. However, the average value from 5 values was used for one replication. Therefore, n=3 is still reported in the table footnote.
6. The histological images in Figure 5 presented as a cross-cut or a longitudinal cut, there was no uniformity. Besides, the different colored arrows in Figure 5 should indicate this.
**** Authors would like to apologize since the detailed information on the planes of sectioning histology was not well described. We used the standard sectioning planes of transversal which are mainly use to cut and describe the tissue structures under microscopy. For this plane, we observed three-orientation of muscle fibers including, longitudinal, oblique and transverse. However, only representative picture of longitudinal to oblique orientation was selected and illustrated in the Figure 5. We chose this orientation because it was easier to assess the sizes and detail of the muscle fiber and other peripheral tissue. For the structural changes, we used different colored arrows to indicate the individual change. We have added the term of orientation in the figure legend for better understanding.
7. 2% COS-BH Day 0 shows a surge in bending vibration of N-H bonds? Does this mean 2% COS-BH protein on day 0? Please explain.
**** Thank you for the meaningful question. The upsurge in the N-H bonds is due to the influence of the exposure of the meat to PEF, causing the modification in the secondary structure at the molecular level, in which the upsurge in N-H bonds was noted. Moreover, the change in the N-H bonds at the molecular level could be visible through the FTIR analysis.
8. It is recommended to put the microbiome load at the top to explain why it is only stored for 3 days (from 3.7 to 3.1).
**** Thank you for the suggestion. Unfortunately, the change in microbiome load would not be possible in the present study. The major objective of this study was to maintain the textural property, which was prone to softening due to the action of proteases, mainly trypsin. The interrelation between TCA soluble peptide, protein pattern, and shear force are directly discussed along with the microbiome load. Thus, microbial growth was the consequence of the plenty of nutrients mainly from the protein hydrolysis. The flow of the content fits well with the objective of this study. Hence, we may not be able to implement the recommended changes.
9. PEF pretreatment combination containing 2%COS delayed the autolysis process of HMS meat. The effect of COS can be determined by comparing different concentrations, but there is no comparison of PEF pretreatment, so how do you confirm its effect?
**** Thank you for the invaluable question. The authors would like to mention that the role of PEF was to create pores on the hard shells of the HMS and consequently allow the COS solutions to penetrate into the muscle of the HMS.
Based on our preliminary study for PEF treatment, there was no significant change in protease activity between meat from HMS without and with PEF treatment. Therefore, we excluded those without PEF treatment from the study. Our major objective was to reduce the activity of trypsin by beheading and COS treatment. PEF was the potential means to make a tiny pore for penetration of COS solution.
Reviewer 3 Report
Comments and Suggestions for Authors
The authors of the paper " Impact of Prior Pulse Electric Field and Chitooligosaccharide treatment on Trypsin Activity and Quality Changes of Whole and Beheaded Harpiosquillid Mantis Shrimp during Storage in Iced Water " present a relevant topic, namely "combination between prior PEF and 2% COS treatment of beheaded HMS could effectively maintain quality of HMS stored in iced water", having a primarily applied contribution at the level of microbiological analyses, thus appreciating the multiplying effect of the work. However, we would appreciate if from the abstract of the work the authors could mention the scientific contribution and the innovative elements resulting from the research and which have multiplying effects, given the fact that the impact of this study on microbiological analysis.
Concepts are properly mentioned in the paper, as well as bibliographic sources and citations are properly mentioned in the paper, for example the authors use bibliographic references, such as "loss of quality caused by proteolysis and microorganisms, and also extended the shelf life of shrimps [11,12]". We recommend, however, the introduction of a separate "literature review" chapter to support the academic accuracy of the authors' scientific documentation.
In the framework of the research methodology, the authors start in their academic endeavors, from the justification of the chemical products used in the research, respectively "chemical substances of analytical quality were purchased from Sigma Aldrich (St Louis, MO, UNITED STATES OF AMERICA). All media for microbial assays were procured from Hi-Media Laboratories (Mumbai, India).” Moreover, the authors demonstrate "trypsin activity", according to "the procedure of Khantahant and Benjakul [25] using BAPNA as substrate at pH 8.0 and 60℃ for 20 minutes", and according to the mathematical formula, which shows us a structuring appropriateness of the research methodology by the authors of the work.
The results of the study are presented both descriptively and graphically in the form of figures and tables presented by the authors, highlighting the fact that "various bacteria play a key role in the spoilage of seafood and their enzyme products causing HMS proteolysis during storage". The presented results are pragmatically oriented towards applied research, but we suggest the authors of the papers to present the innovations and personal scientific contributions developed within the paper and which contribute directly to the specialized scientific literature.
The conclusions are presented by the authors of the paper highlighting the fact that " PEF pretreatment in combination with 2% COS of beheaded HMS could lower trypsin activity in HMS meat and delayed the autolysis process.". At the same time, we suggest the authors of the paper to highlight as clearly as possible the limitations of the study, as well as future research with multiplying effects.
We congratulate the research team for the theme of the paper carried out, we suggest revising the work according to the recommendations maintained within the work and thank you.
Author Response
Reviewer 3
****Thank you so much for the comment and suggestion. All queries have been responded and the corrections have been done as highlighted in green.
The authors of the paper " Impact of Prior Pulse Electric Field and Chitooligosaccharide treatment on Trypsin Activity and Quality Changes of Whole and Beheaded Harpiosquillid Mantis Shrimp during Storage in Iced Water " present a relevant topic, namely "combination between prior PEF and 2% COS treatment of beheaded HMS could effectively maintain quality of HMS stored in iced water", having a primarily applied contribution at the level of microbiological analyses, thus appreciating the multiplying effect of the work. However, we would appreciate if from the abstract of the work the authors could mention the scientific contribution and the innovative elements resulting from the research and which have multiplying effects, given the fact that the impact of this study on microbiological analysis.
**** Thank you for your invaluable suggestion and time spent throughout the manuscript. Following the reviewer’s suggestions, the modifications have been made in the abstract. The synergetic effect of combined treatment of beheading, PEF and chitooligosaccharides (COS) on the retardation of microbial growth has been included. Please see line number 33-35.
Concepts are properly mentioned in the paper, as well as bibliographic sources and citations are properly mentioned in the paper, for example the authors use bibliographic references, such as "loss of quality caused by proteolysis and microorganisms, and also extended the shelf life of shrimps [11,12]". We recommend, however, the introduction of a separate "literature review" chapter to support the academic accuracy of the authors' scientific documentation.
**** Thank you for the invaluable compliment. The reviewer’s suggestion given has been taken for consideration in our future work. However, authors prefer to keep the introduction including the literature in the present form, in which the technology used (PEF) or COS were separately presented into different paragraphs for better understanding. Problem and cause of softening was also addressed in the first paragraph.
In the framework of the research methodology, the authors start in their academic endeavors, from the justification of the chemical products used in the research, respectively "chemical substances of analytical quality were purchased from Sigma Aldrich (St Louis, MO, UNITED STATES OF AMERICA). All media for microbial assays were procured from Hi-Media Laboratories (Mumbai, India).” Moreover, the authors demonstrate "trypsin activity", according to "the procedure of Khantahant and Benjakul [25] using BAPNA as substrate at pH 8.0 and 60℃ for 20 minutes", and according to the mathematical formula, which shows us a structuring appropriateness of the research methodology by the authors of the work.
**** Thank you for the compliment on the research methodology. The authors appreciate it with gratitude.
The results of the study are presented both descriptively and graphically in the form of figures and tables presented by the authors, highlighting the fact that "various bacteria play a key role in the spoilage of seafood and their enzyme products causing HMS proteolysis during storage". The presented results are pragmatically oriented towards applied research, but we suggest the authors of the papers to present the innovations and personal scientific contributions developed within the paper and which contribute directly to the specialized scientific literature.
**** Thank you for the important suggestion. The authors would like to address a major concern of muscle softening in HMS. The combined treatment helped maintain the textural property of HMS muscles during the storage. In the present study, the innovation consisted of the use of non-thermal processing namely PEF. Also, chitooligosaccharide was used at the first time to inactivate the trypsin from HMS. The aforementioned tools contributed to inhibiting the enzymatic proteolysis as well as retardation of microorganism growth, which were the critical cause for softening of HMS.
The conclusions are presented by the authors of the paper highlighting the fact that " PEF pretreatment in combination with 2% COS of beheaded HMS could lower trypsin activity in HMS meat and delayed the autolysis process.". At the same time, we suggest the authors of the paper to highlight as clearly as possible the limitations of the study, as well as future research with multiplying effects.
**** Thank you for the invaluable suggestion. The conclusion has been revised per the reviewer's suggestion, highlighting facts, limitations, and future research. Please see line numbers 564-568.
We congratulate the research team for the theme of the paper carried out, we suggest revising the work according to the recommendations maintained within the work and thank you.
**** Authors would be grateful to reviewers for valuable suggestions and comments.